# Bifurcations and Proarrhythmic Behaviors in Cardiac Electrical Excitations

**DOI:** 10.3390/biom12030459

**Published:** 2022-03-16

**Authors:** Kunichika Tsumoto, Yasutaka Kurata

**Affiliations:** Department of Physiology II, Kanazawa Medical University, Uchinada 920-0293, Japan; tsumoto@kanazawa-med.ac.jp

**Keywords:** action potential, nonlinear dynamical system, bifurcation theory, afterdepolarizations, cardiac arrhythmias

## Abstract

The heart is a hierarchical dynamic system consisting of molecules, cells, and tissues, and acts as a pump for blood circulation. The pumping function depends critically on the preceding electrical activity, and disturbances in the pattern of excitation propagation lead to cardiac arrhythmia and pump failure. Excitation phenomena in cardiomyocytes have been modeled as a nonlinear dynamical system. Because of the nonlinearity of excitation phenomena, the system dynamics could be complex, and various analyses have been performed to understand the complex dynamics. Understanding the mechanisms underlying proarrhythmic responses in the heart is crucial for developing new ways to prevent and control cardiac arrhythmias and resulting contractile dysfunction. When the heart changes to a pathological state over time, the action potential (AP) in cardiomyocytes may also change to a different state in shape and duration, often undergoing a qualitative change in behavior. Such a dynamic change is called bifurcation. In this review, we first summarize the contribution of ion channels and transporters to AP formation and our knowledge of ion-transport molecules, then briefly describe bifurcation theory for nonlinear dynamical systems, and finally detail its recent progress, focusing on the research that attempts to understand the developing mechanisms of abnormal excitations in cardiomyocytes from the perspective of bifurcation phenomena.

## 1. Introduction

We can observe a wide variety of dynamics in life phenomena. Dynamics are the changes in states of a system over time. The system is defined as a set of objects with functions; if the states of the system change with time, then the system is called a dynamic system. A mathematical model is a description of the rule for state changes in the system, and such a model is called a dynamical system [1,2].

The heart is a highly hierarchical dynamic system consisting of molecules, cells, and tissues [3,4,5]. It is well known that cardiac muscle cells cause an electrical excitation. To reproduce the excitation phenomena observed in cardiac cells, many mathematical models have been proposed. Excitation is a nonlinear phenomenon; the excitatory dynamics of cardiomyocytes modeled as a nonlinear dynamical system could be complex. For understanding complicated dynamics, various methods based on nonlinear dynamical system theories have been utilized.

Electrical phenomena in the heart are initiated with the spontaneous excitation evoked in pacemaking cells in the sinoatrial (SA) node, followed by the excitation conduction to the atrium, atrioventricular (AV) node, His bundles, Purkinje fiber, and ventricle [6]. By regularly repeating the precise and coordinated excitation propagation, the heart plays an essential role as a circulating blood pump. Cardiac arrhythmias are disturbed heartbeats or uncoordinated excitation propagations.

In the 1980s–1990s, the idea of dynamical diseases was proposed by some researchers [7,8,9]. The idea assumes the existence of a certain kind of nonlinear dynamical system behind phenomena evoked in the heart and then links the normal function of the heart, i.e., normal heartbeat, to a stable state of the nonlinear dynamical system. This will allow us to consider the regular and coordinated excitation propagation that underlies the normal heartbeat as one of the desirable stable states for the cardiac system. Furthermore, the phenomenon that cardiac excitations and/or propagations are disturbed by ischemia, stress, drug action, etc., which is called *arrhythmia*, can be considered as the dynamical disease. In other words, cardiac arrhythmias can be captured as state changes in the cardiac system, with the desirable stable state being changed into another undesirable one by changing system parameters in the heart related to many factors.

Understanding how the proarrhythmic response occurs in the heart is extremely important for developing new ways to prevent and control cardiac arrhythmia and contractile dysfunction. Based on the concept of dynamical disease, a change in some of the system components may cause a breakdown in the electrical excitation of cardiomyocytes; this is a transition from a normal stable state to another stable state. Such a state change, a qualitative change in system behaviors, or a sudden change of phenomenon, is called a bifurcation (or bifurcation phenomenon) in nonlinear dynamical systems theory [10,11,12,13]. Bifurcation analysis is the search for system parameters whose changes abruptly alter system behaviors. This review summarizes recent progress in understanding the mechanisms of normal and abnormal excitations observed in cardiomyocytes from the perspective of bifurcation phenomena.

## 2. Ionic Mechanisms of Excitations in Cardiomyocytes

Working cardiac muscles such as atrial and ventricular myocytes are excitable cells (excitable membrane) and are quiescent at a constant potential called “resting membrane potential” (−80 to −90 mV) in the absence of an external stimulus. The membrane potential of atrial and ventricular myocytes changes transiently upon the input of an appropriate external stimulus that depolarizes the membrane potential above a threshold level. This membrane potential change is referred to as an action potential (AP). On the other hand, specialized cardiac muscles such as sinoatrial node and atrioventricular node cells do not have a constant resting membrane potential, but spontaneously and periodically generate APs. In both cases, membrane potential changes are caused by the movement of ions into and out of the cell through ion-transport molecules such as ion channels and transporters embedded in the cell membrane.

To understand how the proarrhythmic behavior of cardiomyocytes occurs, it is necessary to understand the mechanism of normal AP generation. To achieve this, the ionic contribution to APs should also be understood. In the following, we first summarize ion homeostasis in cardiomyocytes and the mechanisms of AP generation. The biophysical mechanisms of abnormal APs related to arrhythmogenesis will then be summarized.

### 2.1. Mechanisms for Maintenance of Ion Homeostasis

The electrical activity of the heart, e.g., AP generation, is regulated by a variety of ion-transport proteins such as ion channels and ion transporters/exchangers embedded in the cell membrane. In cardiomyocytes, the intra- and extracellular environments are separated by a lipid bilayer membrane, and there are differences in ion concentrations. The major ions that contribute to AP generations in cardiomyocytes are Na^+^, K^+^, and Ca^2+^, and the extracellular concentrations of Na^+^ and Ca^2+^ are higher than their intracellular concentrations. For example, when Na^+^ or Ca^2+^ channels are opened within the range of physiological membrane potentials, the ions move from the outside to inside the cell by diffusion according to an electrochemical gradient, increasing their intracellular concentrations. On the other hand, the intracellular K^+^ concentration is higher than the extracellular K^+^ concentration. Thus, the opening of K^+^ channels results in the outflow of K^+^ ions from inside to outside the cell. When only K^+^ channels remain open, the membrane potential will reach the reversal potential of the K^+^ channel, i.e., the equilibrium potential for K^+^. If all types of ion channels, Na^+^, K^+^, and Ca^2+^ channels, continue to open, then the concentration of each ion in the intracellular solution will reach that of the extracellular solution because of the thermodynamic principle. However, the intracellular ion concentrations are always maintained within certain ranges different from extracellular ones. In particular, the intracellular Ca^2+^ regulation (or Ca^2+^ handling) in cardiomyocytes is not only crucial for muscle contraction and relaxation but is also involved in the regulation of ion channel function through Ca^2+^-dependent signaling as a second messenger [14,15,16]. Furthermore, intracellular Ca^2+^ handling is also involved in ion channel remodeling via changes in the activity of transcription factors. Ca^2+^-handling abnormalities due to disruption of intracellular Ca^2+^ ion homeostasis, such as those in ischemia, hypertrophy, and heart failure, are a major cause of the development of lethal arrhythmias (for details, see reviews in [17,18]). In cardiomyocytes, three transporters, Na^+^/K^+^ ATPase (NKA) [19,20,21,22,23], Na^+^-Ca^2+^ exchanger (NCX) [24,25,26], and Na^+^-H^+^ exchanger (NHE) [27], play a key role in maintaining ion homeostasis. NKA and NCX are electrogenic because they generate ionic currents, while NHE is electroneutral. Therefore, it is NKA and NCX that are directly involved in AP generation in cardiomyocytes.

NKA hydrolyzes ATP to generate chemical energy. The generated energy is used to exchange three intracellular Na^+^ ions and two extracellular K^+^ ions against each electrochemical gradient. When NKA works once, one extra cation is released from inside to outside the cell. Therefore, the NKA current (*I*_NKA_) is an outward current, which can shorten the AP and hyperpolarize the resting membrane potential. So far, four isoforms of the α subunit, α1, α2, α3, and α4, are found to be expressed in mammals and three isoforms of the β subunit, β1, β2, and β3, have been identified [28,29]. The α1, α2, α3, α4, β1, β2, and β3-subunits are encoded by the genes *ATP*1*A*1, *ATP*1*A*2, *ATP*1*A*3, *ATP*1*A*4, *ATP*1*B*1, *ATP*1*B*2, and *ATP*1*B*3, respectively [30]. In rodents, α1 and α2 were the two major isoforms [31], while in dogs and macaques, α1 and α3 were mainly expressed [32]. In the human heart, three α isoforms, α1, α2, and α3, have been detected [32], and their expressions have been estimated to be 62%, 15%, and 23%, respectively [33,34]. In particular, the α1-subunit is widely distributed in the surface sarcolemma [24] and plays a major role in regulating Na^+^ ion concentrations in the cytoplasm. The α2-subunit is mainly expressed in the dyadic cleft and may play a special role in regulating the transport of Ca^2+^ released from the sarcoplasmic reticulum via interaction with NCX [22]. The β-subunit is also essential for the pump function and plays a role in the α-subunit stabilization.

The main function of NCX is to exchange three extracellular Na^+^ and one intracellular Ca^2+^ [35,36,37]. At this time, the electrochemical potential energy of Na^+^ that is lost during its transport is used to transport Ca^2+^ against its electrochemical gradient. At the resting membrane potential in ventricular or atrial myocytes (~−80 mV), the ion Ca^2+^ is pumped out of myocytes by NCX. Furthermore, NCX generates an ionic current (*I*_NCX_) inwardly for most of the time during AP generation. Thus, *I*_NCX_ can contribute to AP prolongations and depolarize the resting potential in working myocytes. Note that NCX can generate outward currents in the late phase of rapid AP depolarization where the membrane potential overshoots before an increase in the intracellular Ca^2+^ concentration. *I*_NCX_ turns outward in the late depolarization phase because the membrane potential becomes more depolarized than the reversal potential of NCX. This is called the “reverse mode” of NCX, causing Ca^2+^ influx into the cell. The reverse mode leads to increased intracellular Ca^2+^ concentrations, playing a role in the augmentation of the muscle contraction via the excitation–contraction coupling.

NCX is a member of the huge Ca^2+^/cation antiporter superfamily [26]. Mammalian cells express three NCX isoforms, NCX1, NCX2, and NCX3, which are encoded by the genes *SLC*8*A*1, *SLC*8*A*2, and *SLC*8*A*3, respectively [38,39]. The representative of the cardiac NCX isoform is NCX1.1 [35].

### 2.2. Ionic Mechanisms of Cardiac Action Potentials

The transmembrane potential in cardiomyocytes changes by approximately 60 to 120 mV with AP generation. APs initiate from the resting membrane potential in working myocytes (~−80 mV) or the maximum diastolic potential in specialized cardiomyocytes (~−40 mV) that constitute the conduction system and exhibit automaticity. In working myocytes, the resting membrane potential is mainly influenced by the inwardly rectifying K^+^ channel current (*I*_K1_), which is mediated by the inward-rectifier K^+^ channel (Kir channel) consisting of the Kir2.1 subunit encoded by *KCNJ*2 [40,41]. The *I*_K1_ also functions to terminate the AP and to indirectly determine the excitability of cardiomyocytes. Furthermore, the NKA contributes to the maintenance of the resting membrane potential [23].

APs (time evolution of the membrane potential) of cardiomyocytes differ in shape in different parts of the heart, and also vary greatly between species. Here, we consider the AP of human ventricular myocytes (see Figure 1).

By applying appropriate stimulation to ventricular myocytes in a resting state, the cell membrane is depolarized. Depolarization of the membrane potential first activates Na^+^ channels [42], which mainly consist of the Na_V_1.5α subunit [43] encoded by *SCN*5*A*, causing Na^+^ to flow into the cardiomyocyte. The influx of cations into the cell further depolarizes the membrane potential, leading to further activation of Na^+^ channels. As a result, more Na^+^ ions flow into the cell, and depolarization is accelerated. The influx of Na^+^ into the myocyte in a positive-feedback manner causes a very large inward Na^+^ channel current (*I*_Na_) and rapidly depolarizes the membrane potential to form AP phase 0. However, the inactivation of Na^+^ channels is also rapid, and *I*_Na_ will soon stop flowing.

Depolarization of the membrane potential also activates other ion channels such as K^+^ and Ca^2+^ channels. Transient outward K^+^ channels, which are mainly composed of K_v_4.3 and K_v_4.2 α-subunits encoded by *KCND*3 and *KCND*2, respectively, in human and canine ventricular myocytes [44,45], are activated following AP phase 0 and produce a large transient outward K^+^ current (*I*_to_) [46]. As a result, the membrane potential transiently repolarizes to form AP phase 1. In human cardiomyocytes, the K_v_1.4 α-subunit is also expressed, making up 10–20% of the *I*_to_ density [47,48].

Following the AP overshoot, Ca^2+^ channels [49] and delayed rectifier K^+^ channels [50] are activated to form AP phase 2. There are two major subtypes of Ca^2+^ channels expressed in cardiomyocytes: L-type Ca^2+^ channels (LTCCs) and T-type Ca^2+^ channels [51,52]. The most abundant type expressed in ventricular myocytes is the LTCC (CaV1.2), which carries Ca^2+^ currents through a pore-forming α-subunit (α1) encoded by the *CACNA1C* gene [46]. T-type Ca^2+^ channels (CaV3.1 and CaV3.2), encoded by the *CACNA1G* and *CACNA1H* genes, respectively [53,54], are believed to be involved in the automaticity of the sinoatrial node cell. However, their functional role in ventricular myocytes is not clear. Delayed rectifier K^+^ channel currents, carrying K^+^ out of the myocyte, can be separated into a rapid component (*I*_Kr_) and a slow component (*I*_Ks_) [50]. The *I*_Kr_ channel α-subunit is encoded by a human *ether-à-go-go-related gene* (hERG) and is also called the hERG channel and Kv11.1 pore-forming α-subunit [55,56,57]. In addition, *I*_Ks_ is thought to be carried by the Kv7.1 channel encoded by the *KCNQ1* gene [58,59,60,61]. Activation of the LTCC is sustained for a relatively long time and results in a slow inward current (*I*_CaL_). On the other hand, *I*_Kr_ and *I*_Ks_ carry outward currents, though there is only a small current flow of *I*_Kr_ during AP phase 2 as hERG channels rapidly enter into the inactive state [62,63]. There is a certain balance between the inward Ca^2+^ and outward K^+^ currents during this phase. This balance forms the plateau phase (AP phase 2) in AP of the ventricular myocyte.

Eventually, LTCCs also become inactive due to the two mechanisms, voltage- and Ca^2+^-dependent inactivation, and *I*_CaL_ (inward current) becomes smaller. In the subsequent phase (AP phase 3), the outflow of cations by *I*_Kr_ and *I*_Ks_ causes the membrane potential to rapidly repolarize. Particularly for *I*_Kr_, as the membrane potential starts to repolarize, hERG channels recover from inactivation and carry more *I*_Kr_. These larger outward currents (*I*_Kr_ and *I*_Ks_) contribute to the increased rate of repolarization. With the large flow of *I*_K1_ in the latter half of AP phase 3, the AP undergoes a transition from the repolarization phase to the resting state (AP phase 4). After that, *I*_K1_ channels continue to discharge K^+^ out of the cardiomyocyte, consequently flowing the outward current and maintaining a deep resting membrane potential; NKA counteracts *I*_K1_ channels, returning K^+^ to the intracellular space. The *I*_K1_ channels consist of the Kir2.1, Kir2.2, Kir2.3, and Kir2.4 pore-forming α-subunits, which are encoded by the *KCNJ2*, *KCNJ12*, *KCNJ4*, and *KCNJ14* genes, respectively [40,41,64,65].

### 2.3. Mechanisms of Abnormal Action Potentials (Afterdepolarization)

APD prolongation in ventricular myocytes increases the risk of the development of lethal ventricular arrhythmias such as torsades des pointes (TdP), often leading to cardiac sudden death in long QT syndrome (LQTS) [66,67,68,69]. At present, inherited LQTS, which is characterized by a congenital prolongation of QT interval, is subdivided into 17 different types depending on where genetic mutations occur (LQT1~LQT17) [66,67,68,69]. For instance, in patients with LQT1 and LQT2, the function of the ion channel responsible for *I*_Ks_ (LQT1) or *I*_Kr_ (LQT2) is reduced by loss-of-function mutations of the K^+^ channels. In both cases, a decrease in repolarization currents delays AP repolarization and prolongs the QT interval. A genetic abnormality of Na_V_1.5 (LQT3) that causes persistent inward Na^+^ currents due to impaired Na^+^ channel inactivation has also been observed; this can be called a gain-of-function mutation of the Na^+^ channel. In the different genotypes, TdP may be precipitated by physical or emotional stress (LQT1), a startle (LQT2), or may occur at rest or during sleep (LQT3). These three types of LQTS account for more than 70% of all inherited LQTS (genetic abnormalities of LQT1, LQT2, and LQT3 account for 30~35%, 20~25%, and 5~10%, respectively) [68].

Normally, the AP of cardiomyocytes is completed by sufficient K^+^ current flow during the repolarization phase (late AP phase 2 and phase 3) of the AP (Figure 1). The presence of multiple K^+^ current components such as *I*_Ks_, *I*_Kr_, and *I*_NKA_, all of which act to repolarize the AP, appears redundant as a system. However, the redundancy forms an available reserve for AP repolarization. This concept is referred to as the “repolarization reserve” [70,71]. If these outward K^+^ channel currents, e.g., *I*_Kr_ and/or *I*_Ks_, attenuate due to genetic abnormalities, drug actions, aging, etc., or if *I*_Na_ and/or *I*_CaL_ continue to flow during the plateau phase, then the repolarization is delayed, leading to excessive AP prolongation. In particular, the continuation of the plateau phase where the membrane potential is within the range of the *I*_CaL_ window current can lead to an excessive accumulation of Ca^2+^ in the cytoplasm. In this situation, abnormal behaviors called afterdepolarizations are likely to occur [70,72,73]. Afterdepolarizations include early afterdepolarization (EAD), which is the transient depolarization during the late AP phase 2 and phase 3, and delayed afterdepolarization (DAD) [74,75,76], which is the transient depolarization after AP completion, i.e., in AP phase 4. Many experimental studies have suggested that the reactivation of the LTCC current (*I*_CaL_) during AP phase 2 and phase 3 in ventricular myocytes is a key mechanism of EAD formation [77,78,79,80,81,82]. Ca^2+^ is taken up into the cardiomyocyte by the Ca^2+^ influx through the LTCC. The LTCC activates at the membrane potential of approximately −30~+10 mV, resulting in a relatively large inward current. On the one hand, the LTCC exhibits a voltage-dependent inactivation (VDI) at the membrane potential of approximately −50~0 mV. The LTCC inactivates during AP phase 2 and Ca^2+^ influx terminates. Thus, during AP repolarization, *I*_CaL_ by LTCCs flows within a range of the membrane potential where the steady-state activation and inactivation curve of the LTCC overlap (see Figure 2), known as “the *I*_CaL_ window current region” [83,84]. AP repolarization results in recovery from the inactivation state of LTCCs. As a result, the LTCC slightly reactivates in the AP repolarization process and causes a small inward current. The repolarization of a normal AP proceeds rapidly, so the period of time during which the membrane potential is in the *I*_CaL_ window current range is short. The time for reactivation of LTCCs is very short, and sufficient current to cause EAD does not occur. However, when the repolarization reserve is reduced, i.e., *I*_Kr_ or *I*_Ks_ or both are reduced, the time at which the membrane potential is in the *I*_CaL_ window current range lengthens. For this reason, *I*_CaL_ can be augmented by LTCC reactivation. At this time, an increase in the inward current component during the AP repolarization phase causes the inward/outward current imbalance in the net membrane current (*I*_net_). Since *I*_K1_ is not yet activated during the late AP phase 2 and early phase 3, the membrane resistance remains relatively high during these phases. This means that even a slight current causes a large change in the membrane potential. As a result, the first EAD is triggered in the late AP phase 2 to phase 3. As modulators of AP with EADs, at least, intracellular Ca^2+^-handling (or SR Ca^2+^-handling), intracellular Na^+^-handling, and the heart rate can be considered [85] (Figure 3). For example, inhibition of NCX (e.g., by SEA-0400 administration [81]) primarily suppresses the Ca^2+^ efflux. As a result, the increase in Ca^2+^ concentrations in the intracellular subspace ([Ca^2+^]_ss_) and the cytoplasm ([Ca^2+^]_i_) promotes the Ca^2+^-dependent inactivation(CDI) of LTCCs and secondarily decreases *I*_CaL_, consequently suppressing EAD formation [86,87]. Conversely, enhanced NCX (e.g., by overexpression [82]) increases Ca^2+^ extrusion from the intracellular to extracellular space, resulting in a decrease in [Ca^2+^]_i_ (or [Ca^2+^]_ss_). The decrease in [Ca^2+^]_i_ (or [Ca^2+^]_ss_) suppresses the CDI of LTCCs and leads to an increase in the *I*_CaL_ window current, resulting in the promotion of EAD formation [88]. As exemplified by the administration of digitalis, NKA inhibition results in a decrease in outward current, i.e., repolarization reserve attenuation, and thus contributes to facilitating EAD generation. As suggested by Xie et al. [89], the membrane potential change via *I*_CaL_, *I*_NCX_, and *I*_NKA_, and the interactions between intracellular Ca^2+^- and Na^+^-handling, influence the development of EADs. Another possible mechanism of EAD formation is spontaneous Ca^2+^ release from the sarcoplasmic reticulum (SR) [90], which will increase the Ca^2+^ concentration in the cytoplasm. The spontaneous SR Ca^2+^ release is also the recognized mechanism of DADs; it has been reported that DADs are often accompanied by EADs [91,92,93]. As a result, the outflow of Ca^2+^ by NCX and inward *I*_NCX_ are enhanced and may cause transient depolarization of the membrane potential. For instance, the decrease in Na^+^ efflux due to chronic inhibition of NKA causes an increase in the intracellular Na^+^ concentration ([Na^+^]_i_). This adversely affects the activity of NCX, which operates by a loss of electrochemical energy during the transfer of Na^+^ from outside to inside the cell. Consequently, increased [Ca^2+^]_i_ and [Ca^2+^]_ss_, and excessive Ca^2+^ accumulation in the SR ([Ca^2+^]_SR_) cause spontaneous Ca^2+^ release from the SR to the cytoplasm, which in turn facilitates EAD generation via NCX activation. Spontaneous SR Ca^2+^ releases have been suggested to induce EADs under certain conditions such as β-adrenergic stimulation via enhancing inward *I*_NCX_ [94,95]. This mechanism for EAD formation may be applicable to the EAD in the late AP phase 3 observed under β-adrenergic stimulation in an LQTS patient [92].

## 3. Mathematical Cardiomyocyte Models and Bifurcation Phenomena

Mathematical modeling of the electrophysiological properties of cardiac cells originated from the first AP model of squid giant axon based on experimental measurements by Hodgkin and Huxley in the 1950s [96]. Since a cardiac cell model in mammalian Purkinje fiber, employing this Hodgkin–Huxley (H–H) formalism, was first developed by Noble [97], mathematical models of heart muscle cells have been refined by incorporating diverse electrophysiological data. At present, cardiomyocyte models have reflected not only differences in cell types but also differences in experimental animal species, e.g., guinea pigs [98,99,100,101], rabbits [102,103], mice [104,105,106,107,108], dogs [109,110,111], and humans [112,113,114,115,116,117,118]. However, there are no substantial differences in the basic structures of mathematical models that reproduce electrophysiological properties of cardiomyocytes (see Figure 4A). For details on electrophysiological and mathematical models of heart muscle cells, refer to the original articles. Furthermore, review articles written by Noble [119] and Amuzescu et al. [120] detail the history of changes in their models.

### 3.1. Mathematical Models of Action Potential in Cardiomyocytes

The basic idea of the H–H formalism is just to consider the cell membrane as a simple electric circuit (Figure 4B). The capacitive property of the cell membrane is represented as the capacitor with a certain capacitance in the electric circuit. Na^+^, K^+^, and Ca^2+^ channels are modeled as the conductors that have conductances denoted as *g*_Na_, *g*_K_, and *g*_Ca_, respectively, and batteries with the electromotive forces *E*_Na_, *E*_K_, and *E*_Ca_, respectively. The electromotive forces of these batteries correspond to reversal potentials of each ion channel, representing ionic movements depending on ion concentration gradients across the cell membrane. In addition to these ion channels, ATP-driven ion pumps that carry ions against the concentration gradients, ion exchangers that exchange ions inside and outside the cell membrane, and other ion transporters also exist on the cell membrane of cardiomyocytes and generate currents. So, the membrane potential dynamics of cardiac cells are simply expressed by differential equations of the electric circuit shown in Figure 4B, generally described as a nonlinear ordinary differential equation as follows:d*V*_m_/d*t* = −(*I*_ion_ + *I*_sitm_)/*C*_m_,(1)
where *V*_m_ is the membrane potential, *C*_m_ is the membrane capacitance, and *I*_stim_ is the external stimulation current that can be a constant, a time-dependent function, or a single short pulse. *I*_ion_ is the net ionic current through the ion channels, transporters, and pumps, i.e., the sum of individual currents mediated by Na^+^, K^+^, and Ca^2+^ channels, NKA, NCX, the Ca^2+^ pump, etc., and it is denoted as follows:*I*_ion_ = *I*_Na_ + *I*_K_ + *I*_Ca_ + *I*_NKA_ + *I*_NCX_ + ⋯.(2)

For example, the fast Na^+^ channel current *I*_Na_ in Equation (2) is denoted by the equation *I*_Na_ = *g*_Na_ × *m*^3^ × *h* × *j* × (*V*_m_ − *E*_Na_), which takes the form of (current) = (conductance) × (voltage), i.e., Ohm’s law [99]. The voltage *E*_Na_ is the Nernst potential or the equilibrium potential of Na^+^. The Nernst potential is the potential where the tendency of ions to move down their concentration gradient is exactly balanced with the force by the electric potential difference; no Na^+^ current flows through the Na^+^ channel when *V*_m_ = *E*_Na_. Furthermore, the term *g*_Na_ × *m*^3^ × *h* × *j* denotes the conductance of Na^+^ channels, where the constant *g*_Na_ is the maximum conductance of the channel and *m*^3^ × *h* × *j* denotes the temporal change of the conductance, that is, the channel open probability. The gating (state) variables *m*, *h*, and *j* take (dimensionless) values between zero and unity and represent the open probabilities of the activation (*m*), fast inactivation (*h*), and slow inactivation (*j*) gates, respectively. Since the variable *m* is an increasing function of *V*_m_ and *h* (or *j*) is the decreasing one, *m* and *h* (or *j*) are also called the activation and inactivation variables, respectively, of the fast Na^+^ channel. The dynamic opening and closing of the gates are described by the following equations:d*z*/d*t* = *α_z_*(*V*_m_)·(1 − *z*) − *β_z_*(*V*_m_)·*z*, or d*z*/d*t* = (*z*^∞^(*V*_m_) − *z*)/*τ*_z_(*V*_m_),(3)
*z*^∞^(*V*_m_) = *α_z_*(*V*_m_)·{*α_z_*(*V*_m_) +*β_z_*(*V*_m_)}^−1^, and *τ*_z_(*V*_m_) = {*α_z_*(*V*_m_) +*β_z_*(*V*_m_)}^−1^(4)
for *z* = *m*, *h*, and *j*, where *α_z_*(*V*_m_) is the rate constant for changing from a closed state to an open state, and *β_z_*(*V*_m_) represents the rate constant for the transition from the open state to the closed state. In addition, *z*^∞^(*V*_m_) is the steady-state open probability to which the gating variable z asymptotically approaches at a given *V*_m_, and *τ*_z_(*V*_m_) is the time constant of relaxation to a steady state. Note that the rate constants *α_z_*(*V*_m_) and *β_z_*(*V*_m_), and the time constant *τ_z_*(*V*_m_) are not constant but dependent on *V*_m_.

It is known that the specialized cells in the conduction system of the heart, represented by SA node cells, spontaneously produce APs, which is the behavior referred to as automaticity. This implies that AP dynamics can be repetitively produced without *I*_stim_ in Equation (1). On the other hand, atrial and ventricular myocytes known as working myocardium are driven by a sinus rhythm originating from the SA node, that is, *I*_stim_ represents a periodic current input that repetitively stimulates the working myocardium at the sinus rhythm (regular rhythm). For instance, we assume that AP responses in a working myocardium model, such as that shown in Equation (1), are evoked by periodic external current stimuli (*I*_stim_). The temporal variations in *I*_stim_ of the rectangular current pulse are expressed as follows:
(5)$$\begin{array}{cc} I_{\mathrm{stim}}(t)\;=\;I_{\mathrm{stim},\max} & \;\;\;\;(0\;\leq \;t\;<\;T_{1}),\\ I_{\mathrm{stim}}(t)\;=\;0 & \;\;\;\;(T_{1}\;\leq \;t\;<\;T), \end{array}$$
where *T* is the pacing cycle length (i.e., period of the current stimuli) and *T*_1_ is the duration for which the stimulus current is sustained at the maximum value (*I*_stim,max_).

Mathematical models for cardiomyocytes, even now, continue to be refined by incorporating biophysical and biochemical processes. Elucidating the mechanism of proarrhythmic behavior in cardiomyocytes using mathematical models does not necessarily mean revealing unknown proarrhythmic factors. Instead, we aim to understand the time evolution of states, i.e., the dynamic phenomena of the cardiomyocyte. The AP evoked in cardiomyocytes is dynamic, and bifurcation analysis has been utilized to understand its dynamical behavior. Again, the purpose of such in silico studies is not only to model new molecular components involved in the development of proarrhythmic responses in cardiomyocytes. For that matter, such new components are often not the cause to be revealed through modeling. For example, ventricular myocytes may exhibit an AP with multiple transient depolarizations (EADs) during repolarization. The cause of EAD development is an excessive prolongation of the APD due to a decrease in repolarizing currents (or persistence of depolarizing currents). However, how much prolongation of the AP causes EAD and how many EADs occur during repolarization would not be able to be determined without analyzing the dynamics of model equations describing temporal changes of membrane potentials in the cardiomyocyte. This is also another motivation for such in silico studies.

Next, the bifurcation theory for nonlinear dynamical systems (bifurcation phenomena) will be briefly explained. One of the characteristics of nonlinear dynamical systems is that a slight change in a system parameter can cause a drastic change in system behavior. Such bifurcation phenomena observed in the cardiomyocyte and the heart will be summarized.

### 3.2. Bifurcations in Nonlinear Dynamical Systems

#### 3.2.1. Definitions of Dynamical Systems

There are various models of dynamical systems. Here, we consider a system with time as an independent variable and with continuous state variables. Mathematical models that describe the system behavior are different depending on whether the time is continuous or discrete. In the former, the system is represented by a set of ordinary differential equations (ODEs). In the latter, it is a set of ordinary difference equations (i.e., recurrence formulas). Even if a mathematical model that we consider is continuous in terms of both states and time and is described by ODEs, it may be treated as a discrete-time system by sampling states at appropriate time intervals to analyze the behaviors of the system.

Now, we suppose that the nonlinear dynamical systems of interest are described by the following equations:d***x***/d*t* = ***f***(***x***, ***λ***),(6)
d***x***/d*t* = ***f***(*t*, *x*, ***λ***),(7)
where *t* is the time, ***x*** = (*x*_1_, *x*_2_, …, *x*_n_)^Tr^, ***λ*** = (*λ*_1_, *λ*_2_, …, *λ*_m_)^Tr^, and ***f*** = (*f*_1_, *f*_2_, …, *f*_n_)^Tr^ are a state vector, parameter vector, and the vector field, respectively, and Tr represents transposition. As shown in Equation (7), a system that explicitly includes time is called a “non-autonomous system”. On the other hand, the system of Equation (6), which does not explicitly include time, is called an “autonomous system”. Behaviors of the system are different depending on whether it is an autonomous or a non-autonomous system, and thus, the methods for analyses are also different. The specialized cardiomyocytes that compose the conduction system in the heart possess automaticity (i.e., exhibit spontaneous and repetitive excitation), and do not require external stimuli for excitation. Therefore, specialized cardiomyocytes such as sinoatrial node and atrioventricular node cells can be modeled as autonomous systems. On the other hand, working myocytes such as atrial and ventricular muscles are typical examples of excitable systems and can be modeled as non-autonomous systems.

One of the characteristics of autonomous systems is that there can exist states where the right-hand-side vector of Equation (6), which defines the state velocity, becomes zero. The state where the velocity becomes zero, i.e., ***f***(***x***, ***λ***) = **0**, is called the “equilibrium point”. The resting membrane potential corresponds to the equilibrium point in the myocardium system.

On the other hand, the non-autonomous system defined by Equation (7) includes time *t* in the state velocity, i.e., the right-hand side of Equation (7). Formally, let us consider the following autonomous system by considering time *t* as a new state variable *τ* (*t* = *τ*) and rewriting Equation (7):d***x***/d*τ* = ***f***(***u***, ***λ***),(8)
d*t*/d*τ* = 1,(9)
where ***u*** = (*t*, ***x***)^Tr^. In this case, since the right-hand side of Equation (9) is unity, then the right-hand side of the autonomous system of Equations (8) and (9) never becomes zero. In other words, there is no equilibrium point in the non-autonomous system of Equation (7). The states of a non-autonomous system flow at a constant speed in the direction of the time axis. For this reason, it is necessary to devise methods of analysis for non-autonomous systems, different from those for autonomous systems.

In this review, we will only consider the case where the vector field of a non-autonomous system is periodic with respect to time, i.e., ***f***(*t* + CL, ***x***, ***λ***) = ***f***(*t*, ***x***, ***λ***), where CL is the period of external stimulus applied to the working myocardium, i.e., the pacing cycle length. Such a system is called a periodic non-autonomous system. Furthermore, states in such a system possess a periodic nature: ***x***(*t* + *k* × CL) = ***x***(*t*), where *k* = 1, 2, …, *m*, which is also referred to as a *k*-periodic solution. The periodic solution corresponds to the periodic oscillation observed in the original system.

Suppose that a cardiomyocyte is repeatedly stimulated at a given CL (Figure 5A). The CL has the following relationship: CL = APD + DI, where DI is the diastolic interval. Now, let us write a relationship between the APD of AP evoked by the *n*th stimulus and that of AP evoked by the next (*n* + 1)th stimulus as
APD*_n_*_+1_ = *P*(APD*_n_*),(10)
for *n* = 0, 1, 2, …, where *P* is a mapping, which is also referred to as the “Poincaré map” [1,121,122,123]. This formula indicates that the APD of AP evoked by the (*n* + 1)th stimulus (APD*_n_*_+1_) can be represented as a function of that of AP evoked by the *n*th stimulus (Figure 5B). In other words, APD*_n_* is mapped to APD*_n_*_+1_ by the mapping *P*. If an AP train behavior evoked in the original non-autonomous system of Equation (7) is periodic and APDs of APs are equal for every stimulus, then APD*_n_* satisfies the following relationship: APD*_n_*_+1_ − *P*(APD*_n_*) = 0. This situation is exemplified in Figure 5B. Such a point in the (APD*_n_*, APD*_n_*_+1_)-plane as shown in Figure 5B is called a “fixed point”. Thus, the periodic response observed in Equation (7) can be in one-to-one correspondence with the fixed point of the Poincaré map *P*. Furthermore, if for some *ℓ* ≠ 1, APD*_n_* = *P^ℓ^*(APD*_n_*), and if all *P^k^*(APD*_n_*), *k* = 1, 2, …, *ℓ* − 1, are different each other, the periodic behavior evoked in Equation (7) corresponds to an *ℓ*-periodic point. This case can also be studied simply by replacing *P* with *P^ℓ^* or as the *ℓ*th iterate of *P*. Usually, however, the explicit form of *P* cannot be obtained. Therefore, the Poincaré map, in general, must be obtained by acquiring the values of state variables for every CL in the numerical simulation.

In the following, we will briefly explain the bifurcation phenomena of equilibrium points and fixed (periodic) points.

#### 3.2.2. Hopf Bifurcation

In autonomous systems, when the Hopf bifurcation is caused by changing the value of a parameter, an oscillatory response (or rhythmic dynamics), called the “limit cycle”, emerges, as illustrated by the thick circle with arrowheads in Figure 6A,B. For instance, cells in the myocardial sleeve of the pulmonary vein do not normally exhibit automaticity [126,127]. However, for some reason, they suddenly acquire spontaneous excitations (automaticity), triggering paroxysmal atrial fibrillation [128]. The phenomenon such that the convergence behavior to a stationary state (an equilibrium point) suddenly switches to oscillatory motion (Figure 6C) is a typical example of the Hopf bifurcation [129].

#### 3.2.3. Saddle-Node Bifurcation 

In the saddle-node bifurcation (or fold Bifurcation [123]) of equilibrium points, changes in parameters cause stable and unstable equilibrium points to coalesce and disappear. Thus, the number of equilibrium points can change (see Figure 7A). On the other hand, there are also saddle-node bifurcation phenomena of periodic solutions in non-autonomous systems (or of limit cycles in autonomous systems). As in the case of equilibrium points, this bifurcation causes a pair of stable and unstable points to disappear or emerge (Figure 7B). In numerical simulations, the states of equilibrium points and/or fixed (periodic) points may jump significantly when the parameter infinitesimally changes from this bifurcation point (this is referred to as a “jump phenomenon”). Moreover, a system that has reached a new state via the saddle-node bifurcation cannot be returned to its original state, even if the value of system parameters is returned to the bifurcation point (this phenomenon is called “hysteresis”). The hysteresis and jump phenomena are associated with the development of a bi- or multi-stable phenomenon of equilibrium points or periodically evoked APs [130,131].

Gadsby and Cranfield illustrated that there exist two stable resting (equilibrium) states in the cardiac Purkinje fiber cell and that the application of a stimulus causes the membrane potential to converge to different resting membrane potentials [132,133]. Jalife and Antzelevitch demonstrated that when a brief depolarizing pulse was applied during SA node pacemaking, spontaneous repetitive APs were annihilated and the membrane potential converged to another stable state; that is, the resting state [134]. These are typical examples that exhibited the bi-stability of two equilibrium states or a periodic response and an equilibrium state. To list other examples of the hysteretic phenomenon, previous experimental studies have investigated AP responses to changes in CL, i.e., rate-dependence and restitution properties of APD, using cardiomyocytes of guinea pigs, rabbits, and canines [124,135,136,137], and found hysteretic responses of APD to CL changes.

Finally, this bifurcation phenomenon is also involved in an entrainment phenomenon. The specialized cells in the conduction system of the heart, represented by SA node cells, spontaneously and periodically produce APs, being typical examples of self-excited oscillatory systems. When such a self-excited system is perturbed by an external periodic stimulus, if the ratio of the oscillation frequency (frequency of the limit cycle) of the self-excited system and the frequency of the external stimulus is close to a simple rational number, the self-excited oscillation may be entrained by the external periodic stimulus. This is a kind of synchronization phenomenon and is also called “entrainment”. For example, a periodic AP generated in the SA node periodically stimulates the atrioventricular (AV) node (of a self-excited system) when it conducts from the atria to the ventricles. At this time, the self-excited oscillation in the AV node is entrained by the excitation frequency of the SA node cells, resulting in a synchronized periodic state. This frequency mismatch that allows the AV node cell to be entrained is limited to a certain frequency range, depending on the stimulus strength. The relationship between the frequency mismatch range, the so-called “phase lock range”, and the stimulus strength is known to show a characteristic structure (called “Arnold’s tongue structure” [123,138]), see Figure 7C. Then, the phase-locked state may disappear with a saddle-node bifurcation when the stimulus frequency or intensity changes. After that, asynchronous (quasi-periodic) rhythms may occur in which two rhythms with different frequencies proceed with each other without much interaction [9,131].

#### 3.2.4. Homoclinic Bifurcation

The aforementioned saddle-node and Hopf bifurcations of the equilibrium point are classified as “local bifurcations” because they are attributed to the stability change of the dynamics near an equilibrium point. On the other hand, there also exist bifurcation phenomena such that the behavior of the solution is developed over a large region within the state space. These bifurcations are called “global bifurcations” because they are related to the global behavior of the solution [123].

Stable (resp. unstable) manifolds are hypersurfaces within the state space formed by sets of initial values approaching (resp. moving away from) an equilibrium point or a limit cycle over time (Figure 8A). When system parameters change, the stable and unstable manifolds of a given saddle equilibrium point may be connected and a closed orbit (also called a “homoclinic orbit”) may emerge. This phenomenon is called “homoclinic bifurcation”, see Figure 8B.

Miake et al. reported that in guinea pigs, the dominant-negative suppression of Kir channels by viral gene transfer could convert ventricular myocytes that exhibit a quiescent state into pacemaker-like cells that generate spontaneous and repetitive APs [139]. The study of mathematical modeling and bifurcation analyses on biological pacemaker activity of ventricular myocytes by Kurata et al. [112] illustrated that the disappearance of a resting state via a saddle-node bifurcation by suppressing *I*_K1_ evoked automaticity in the ventricular myocyte. This change in dynamics seems to involve the occurrence of a saddle-node homoclinic bifurcation [123] or a saddle-node on an invariant cycle (SNIC) bifurcation [140], in which a saddle-node bifurcation and a homoclinic bifurcation at an equilibrium point occur simultaneously (see Figure 8C, and Refs. [123,140] for details and complete definitions).

#### 3.2.5. Period-Doubling Bifurcation 

The stability of a fixed point on the Poincaré section changes on the occurrence of this bifurcation and a periodic orbit with a double period emerges or disappears (see Figure 9A). The period-doubling bifurcation is also called a flip Bifurcation [123]. Typical examples of this bifurcation are the development of AP alternans [141,142,143,144]. When a cardiomyocyte is periodically stimulated at a given CL, the relation of APD and DI to CL is expressed as CL = APD + DI. In general, if the CL is shortened, then both APD and DI are also shortened. As the DI shortens, repolarization currents, especially *I*_Ks_ with relatively slow deactivation kinetics, accumulate with each stimulus; *I*_Ks_ does not have an inactivation mechanism. This implies that shortened DI causes *I*_Ks_ activation by the next AP generation before *I*_Ks_ completely returns to its original state by deactivation. As a result of the accumulation of activated *I*_Ks_ channels, the increased repolarization current accelerates AP repolarization, leading to APD shortening and DI prolongation. The DI prolongation leads to more progress in *I*_Ks_ deactivation and returns more *I*_Ks_ channels to a closed state. Thus, in the AP generated by the next stimulus, the repolarization current decreases, resulting in APD prolongation, i.e., DI shortening. This alternating change in APD, i.e., repeated prolongation and shortening of APD during stimuli, can also be described as follows:APD*_n_* = *P*^2^(APD*_n_*).(11)

This is the phenomenon called “APD alternans”. APD alternans associated with recovery from inactivation of Na^+^ channels has also been reported, albeit in in silico studies [145,146]. Furthermore, the period-doubling bifurcation is known to successively occur, resulting in a chaotic response [142,147]. This implies that SA node pacemaking becomes irregular or arrhythmic [125], i.e., sinus arrhythmias occur via the sequence of period-doubling bifurcations.

#### 3.2.6. Neimark–Sacker Bifurcation

If this bifurcation occurs, quasi-periodic oscillations may emerge or disappear (see Figure 9B). In the state space, the stability of a fixed (or periodic) point on a Poincaré section changes, and a certain point sequence emerges around the fixed (periodic) points. The point sequence in a mapping constitutes a closed-loop such as a limit cycle in an autonomous system. Because of this, it is called an “invariant closed curve (ICC)”. The ICC corresponds to a doubly periodic (quasi-periodic) oscillation of the original periodic non-autonomous system. The Neimark–Sacker bifurcation is also known as a torus bifurcation, or the Hopf bifurcation in discrete systems [123].

To our knowledge, there exists no experimental evidence that changes in dynamics via the Neimark–Sacker bifurcation were observed at a cellular level. On the other hand, an experimental study demonstrated that the reentrant wave became unstable with fluctuating circulatory time and APD, exhibiting a quasi-periodic pattern [148]. The excellent studies by Qu et al. [149,150] reported that the meandering reentrant excitation wave exhibits quasi-periodic dynamics. They focused on tip dynamics of spiral wave reentry. The reentrant excitation wave was periodically sampled and a return mapping of its tip dynamics was constructed; as the conductance of LTCCs was increased, point sequences generated by the return mapping changed from fixed point-like sequences to a closed-loop shape. In the context of bifurcation theory, this closed-loop is a certain type of ICC, and the emergence of ICC implies a dynamics change via the Neimark–Sacker bifurcation.

#### 3.2.7. Bifurcation Analysis

Bifurcation analysis involves the investigation of bifurcation phenomena by obtaining a set of parameter values that cause bifurcation (bifurcation set) and a graph of these sets (bifurcation diagram). Even now, many researchers manually track their bifurcation sets by trial and error. These days, however, we can use powerful computational tools, for instance, AUTO [151,152], XPP-AUT [153], MATCONT [154], etc., to perform bifurcation analyses. Furthermore, a unique bifurcation analysis tool was developed by Kawakami in 1984 [155]. Since then, this tool has continued to be improved by him and his co-workers [156]. Parker and Chua showed a concrete procedure for the numerical calculation of bifurcation sets [157]. We will leave the details on the bifurcation phenomena of equilibrium points and their analyses to other references [2,10,13,152], and recommend relevant literature [1,10,158] as a good source for these topics. For readers who are interested in the bifurcation analysis of dynamical systems, we recommend an excellent monograph by Kuznetsov [123], which summarizes bifurcation phenomena from the standpoint of applications and would clarify the points that could not be explained in this review.

## 4. Bifurcation Analyses of Proarrhythmic Behaviors

Bifurcation analysis examines a sudden change in dynamic behaviors depending on changes in system parameters and enables us to determine the parameter ranges over which the AP response is stable and the transition mechanisms of the AP responses. One expectation for bifurcation analysis may be to know how to control dynamic responses such as APs from the viewpoint of prediction and/or prevention. When abnormal APs in cardiomyocytes occur, we will hope to control their dynamics and restore them to normal APs. However, in general, we have no way of knowing when, where, and how the parameters of a system should be modified to attain the best control and restoration. Bifurcation analysis is a way to gain insight into the fundamental properties of dynamical systems, and thus, may be able to provide helpful clues for controlling the AP response in cardiomyocytes.

### 4.1. Bifurcation Analyses of Cardiac Cell Models

In early studies, primitive mathematical models that assume only myocardial excitability and its refractoriness have been utilized to understand the complex dynamics of APs evoked in cardiomyocytes. Analyses of the AP dynamics evoked in these simple models helped to understand the dynamical mechanisms of the AV conduction block [144,159,160] and the modulation of the SA node pacemaker by the sympathetic and parasympathetic nervous systems [161,162,163,164]. Furthermore, the arrhythmia models of parasystole, in which the ventricle (or atria) is doubly governed by an ectopic pacemaker in addition to the pacemaker rhythm of the SA node, have been analyzed and the generative conditions for complex proarrhythmic dynamics of cardiac excitations, such as bigeminy and trigeminy, have been theoretically clarified [165,166].

As the electrophysiological refinement of cardiomyocyte models has progressed, studies targeting cardiac excitation dynamics of a higher degree of freedom have become mainstream. SA node cell models described as the autonomous system are compatible with existing computational tools such as AUTO, XPP-AUT, and MATCONT for bifurcation analysis [151,152,153,154]. Therefore, SA node cell models have been suitable for bifurcation analyses and have been utilized to elucidate how individual components consisting of ion-transport molecules contribute to normal pacemaking in SA node cells [167,168,169,170,171,172,173,174,175,176]. Bifurcation analyses were also utilized to examine the generation of abnormal oscillations or biological pacemaker activity in ventricular myocytes [177,178,179,180]. On the other hand, bifurcation analyses of periodic non-autonomous systems such as paced atrial and ventricular myocytes [181,182,183] require some kind of ingenuity because it is difficult to apply existing analysis tools to non-autonomous systems without modification. By artificially generating periodic stimuli corresponding to the SA node pacemaking and applying the periodic stimuli to atrial/ventricular myocytes while adjusting the pacing cycle length (PCL), the corresponding APD of the AP train evoked in the cardiomyocyte at every set PCL can be measured experimentally. From the experimentally obtained data, it is possible to derive a one-dimensional discrete dynamical system (e.g., Equation (10)) describing the time evolution of periodic AP trains based on the relationship between the PCL and APD. By investigating bifurcations in the one-dimensional discrete dynamical system, the dynamical mechanisms of abnormal APs observed in the original non-autonomous system (cardiomyocyte with periodic stimuli) have been elucidated. Furthermore, the guinea pig ventricular myocyte model developed by Luo and Rudy (LRd1 model) [184] has often been used for bifurcation analyses of AP dynamics observed in ventricular myocytes [141,185,186,187,188,189]. Although the LRd1 model is classified into a periodic non-autonomous system, investigations of bifurcation phenomena observed in the LRd1 model were first started from a perspective of an autonomous system, that is, analyzing bifurcations of equilibrium points when the amplitude of constant depolarized current stimuli was changed. As a result, various bifurcation phenomena, such as abnormal spontaneous oscillations evoked in ventricular myocytes, have been studied. As will be shown in the next section, the pioneering work of Tran et al., who explained the mechanism of EAD development in terms of bifurcation theory, has also used the LRd1 model [190].

Since the LRd1 model was published in 1991 [184], cardiomyocyte models have become more sophisticated and complex in the past 30 years [101,114,115,116,117,118,191,192,193]. Although intracellular ion concentration changes were not taken into account in the LRd1 model, by integrating intracellular and extracellular ion concentrations into some cardiomyocyte models as state variables, it is currently possible to examine the effects of intracellular Ca^2+^-handling and Na^+^-handling on AP dynamics [194,195,196]. Bifurcation analyses using such cardiomyocyte models with a higher degree of freedom were previously difficult due to low computational power and computing performance, but have recently become possible with improved computer performance [85,130,131,197,198,199]. In the following section, we summarize bifurcation analyses of the AP with EADs observed in more sophisticated cardiomyocyte models.

### 4.2. Bifurcation Analyses of Early Afterdepolarizations

To understand dynamic mechanisms underlying repolarization abnormality, i.e., the development of EAD, the effects of changes in the maximum conductance of various ion channels on APs with EAD observed in aforementioned various ventricular myocyte models have been investigated [81,112,114,117,184,200]. So far, many experimental [201,202,203,204] and theoretical [81,205,206,207,208,209] studies have shown that excessive AP prolongation results in the occurrence of EAD. Assuming decreases in *I*_Ks_ for LQT1 and *I*_Kr_ for LQT2, the LQT1 and LQT2 versions of human ventricular myocyte models, which can reproduce EADs during β-adrenergic stimulation and bradycardia, respectively, have been constructed [85,130,198]. Relationships between EAD occurrence and changes in parameters that are related to factors involved in the genesis and modulation of EAD such as repolarization currents (*I*_Kr_ and/or *I*_Ks_), *I*_CaL_, *I*_NCX_, SR Ca^2+^ pump current (*I*_up_), and heart rate have been comprehensively examined [85,130,198].

How will EAD emergence caused by the LTCC reactivation be explained from the perspective of dynamical system theory? As a strategy to elucidate the generative mechanisms of EADs [85,198], bifurcation analyses and AP simulations were combined (Figure 9a). First, the model of human ventricular myocytes was regarded as an autonomous system, and bifurcation phenomena of equilibrium points and limit cycles that occur in the system were investigated using MATCONT [154] (see Figure 10A, top). Next, AP simulations of the ventricular myocyte model, which is a periodic non-autonomous system, were performed with a fixed set of parameters, and then a phase diagram was constructed by mapping the information of steady-state AP responses (whether an EAD occurred or not) into a parameter plane (or space) (see Figure 10A, bottom). After that, the bifurcation diagram obtained by analyses of the autonomous system and the phase diagram for the non-autonomous system were merged to examine the relationship of bifurcations and dynamic changes in APs that occurred in the original system (Figure 10B). The moderately complex human ventricular myocyte model proposed by Kurata et al. [85,112] (K05) and the more complex models proposed by ten Tusscher et al. [114,115,198] (TP06) and O’Hara et al. [85,117] (ORd) had a common mechanism for EAD development, namely the reactivation of LTCC. Furthermore, EADs distinct in generation mechanism from LTCC reactivation-dependent ones, i.e., those caused by spontaneous Ca^2+^ releases from the SR were identified in the TP06 model [198]. In our previous study [130], whether the dynamic change of APs that causes EAD can be explained as the occurrence of local bifurcations was directly investigated for APs observed in the K05 model as a periodic non-autonomous system without modification (Figure 10C,D). Consequently, it was found that there were parameter regions in which multiple stable AP responses coexisted, and that hysteresis phenomena occurred for the change in the maximum conductance of *I*_Kr_. However, EAD developments could not be simply explained as resulting from an occurrence of the local bifurcation.

A series of analyses advanced the understanding of the dynamical mechanisms of transitions between the different AP responses (APs with and without EAD) of ventricular myocytes due to functional changes in individual ion-transport molecules such as ion channels and transporters.

### 4.3. Slow–Fast Decomposition Analyses of Dynamic AP Responses with EADs

APs observed in ventricular myocytes are representative relaxation oscillations. The ventricular AP is composed of rapid and slow membrane potential changes. This means that the dynamical system describing the electrical behavior of ventricular myocytes is composed of a fast subsystem contributing to the rapid membrane potential changes and a slow subsystem contributing to the slow membrane potential changes [210]. Tran and co-workers examined the AP and EAD behavior observed in the LRd1 model [184] by decomposing the original (full) system into the fast and slow subsystems [190]. In their and many other theoretical studies [211,212,213,214,215], the activation gating variable of *I*_Ks_ was regarded as the slow subsystem or a parameter for the fast subsystem of the ventricular myocyte model. Kurata and co-workers [197] also employed the activation gating variable of *I*_Ks_ in the K05 model [112] for the slow subsystem, and in our study [198], the state variable of the intra-SR Ca^2+^ concentration was employed for the slow subsystem. As an example, *I*_Ks_ in [85,112,130,197] was formulated by the following equations:*I*_Ks_ = *g*_Ks_·*n*^2^·(*V*_m_ − *E*_Ks_),(12)
d*n*/d*t* = (*n*^∞^ − *n*)/*τ_n_*,(13)
where *g*_Ks_ and *E*_Ks_ are the *I*_Ks_ channel maximum conductance and reversal potential of *I*_Ks_, respectively. *n* denotes a state variable representing the activation gate of *I*_Ks_ and *n*^2^ corresponds to the open probability of *I*_Ks_ channels. In human ventricular myocyte models, when APs are generated, state variables such as *V*_m_ change rapidly, while *n* (*n*^2^) changes slowly (Figure 11A,B, left). Considering the difference in response speed in the fast and slow subsystems, the slow dynamics of the state variable *n* can be regarded as a parameter change. Thus, bifurcations of dynamical responses observed in the fast subsystem were investigated as a function of the slow variable *n* (see Figure 11A,B, middle). Such a method is known as “slow-fast decomposition analysis” in the field of nonlinear dynamical system theory [140,210,216]. The slow–fast decomposition analysis can provide a definition of EAD and a clear reason why EADs occur in LQTS.

To explain the dynamical mechanism for the occurrence of EAD [190,197,198], dynamic behaviors (orbits) of the full system are superimposed on the bifurcation diagram of the fast subsystem (see Figure 11A,B, middle). The dynamical mechanism of EAD development in the K05 model was explained as follows [85,112,130,197]: The stimulus current (*I*_stim_) input rapidly depolarized *V*_m_ in the full system and *V*_m_ overshot 0 mV. After that, the *I*_Ks_ channel open probability, *n*^2^, which was the slow subsystem, slowly increased, consequently increasing *I*_Ks_ and repolarizing *V*_m_ (Figure 11A,B, right). With increasing *n*^2^ as a parameter for the fast subsystem, an equilibrium point at depolarized *V*_m_ (qEQ_3_) was destabilized via the Hopf bifurcation (see center panels in Figure 11A,B). When the *I*_Kr_ conductance decreased (as in LQT2; Figure 11B), the Hopf bifurcation point of qEQ_3_ in the fast subsystem shifted significantly toward higher *n*^2^ values (compare center panels in Figure 11A,B). This means that the region of stable qEQ_3_ broadened in the direction of higher *n*^2^, shifting the stable limit sets of the limit-cycle qLC to the right. Then, the trajectory of the full system was attracted to the stable limit sets for qEQ_3_ and qLC in the fast subsystem during delayed repolarization. Such trapping constrained the orbit of the full system to move along the stable manifolds of the qEQ_3_ and the qLC. Therefore, the full system showed oscillation-like behavior during late AP phase 2 to phase 3. Since the stable/unstable limit sets of qLCs were lost by crossing the SN bifurcation point, the increase in *n*^2^ above the SN bifurcation point led to the release of the orbit in the full system trapped in the stable manifold of the qLC. Then, *V*_m_ quickly converged to a stationary state (resting state). This suggests that transient depolarization during AP repolarization, i.e., EAD can be explained as limit cycle-like oscillations occurring in the fast subsystem [197,198]. Thus, EADs can be defined as transient oscillations of a full system trajectory around the stable and unstable qEQ_3_ close to Hopf bifurcation points or in the vicinity of the stable qLC during changes of the slow variable. The occurrence of EADs in LQTS can be attributable to the shift in a Hopf bifurcation point on qEQ_3_ in bifurcation diagrams of the fast subsystem. In recent studies, slow–fast decomposition analyses have been performed by selecting various slow state variables, and attempts have been made to explain the dynamical mechanism underlying EADs [82,217,218,219]. We also recently reported another type of EAD formation distinct from the membrane-dependent mechanism described above, e.g., the development of EAD due to spontaneous Ca^2+^ releases from the SR [198].

## 5. Future Directions

This review outlined fundamental points of dynamical systems and bifurcation phenomena and applications of bifurcation analyses to cardiac systems. The AP response of cardiomyocytes changes dynamically due to various factors. We employed bifurcation theory of the dynamical system to elucidate the mechanism of EAD, motivated by our expectation that the occurrence of EAD, which triggers the onset of fatal arrhythmias, can be associated with changes in the dynamical stability of APs (i.e., bifurcation phenomena).

We could analyze the bifurcation phenomena of AP responses observed in the cardiac cellular system without any approximation. Although the bifurcation analyses were able to make clear the dynamic transition mechanisms of AP responses depending on parameter changes, it was difficult to explain the mechanism of EAD development by local bifurcation alone. In this respect, slow–fast decomposition analysis was effective.

There will be no doubt that a theoretical approach based on bifurcation theory is effective in understanding the mechanisms of dynamic behaviors seen in cardiac systems. Although this review focused on understanding the EAD mechanism, the bifurcation analysis may be useful in understanding the developmental mechanism of DADs, which has been believed to be another triggering mechanism for lethal arrhythmias. However, current mathematical models of cardiac cells are still immature, and it cannot be said that actual complex dynamic responses seen in cardiomyocytes can be sufficiently reproduced by the model cells. In addition, cardiac arrhythmias do not denote abnormal excitation that occurs at the cellular level, but rather indicate abnormal electrical phenomena that occur in multicellular systems at the tissue and organ levels. Thus, we must consider bifurcations of excitation propagation (not AP responses). We previously reported several in silico studies [145,146] investigating the effects of system parameter changes on excitation propagations observed in simple models of myocardial fiber or strand using a brute force method, which is a kind of parameter study, and by constructing phase diagrams such as bifurcation diagrams. However, the existing bifurcation theory alone may be insufficient for analyzing the bifurcation phenomena with high degrees of freedom, such as changes of excitation propagation before and after the development of lethal arrhythmias. It will be important not only to improve model cells by further experimental studies but also to develop new analysis methods that can be applied to multicellular models. In the future, we would like to develop the control theory of cardiac arrhythmias and establish strategies for their treatments.

## Figures and Tables

**Figure 1 biomolecules-12-00459-f001:**
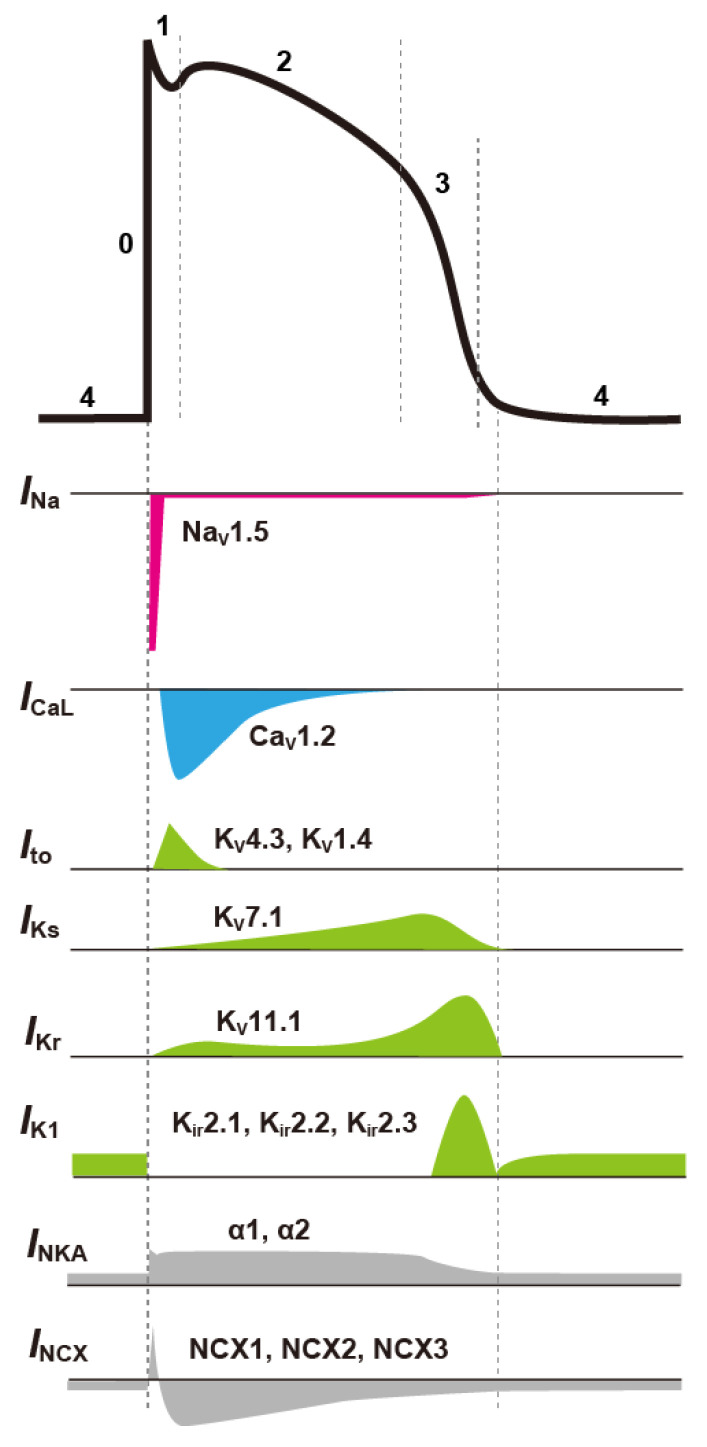
Schematic representation of a typical action potential (AP) in the ventricular myocyte and the respective ion channel currents that contribute to AP formation. It also indicates the major molecules that constitute ion channels and transporters (for the meanings of individual symbols, see text).

**Figure 2 biomolecules-12-00459-f002:**
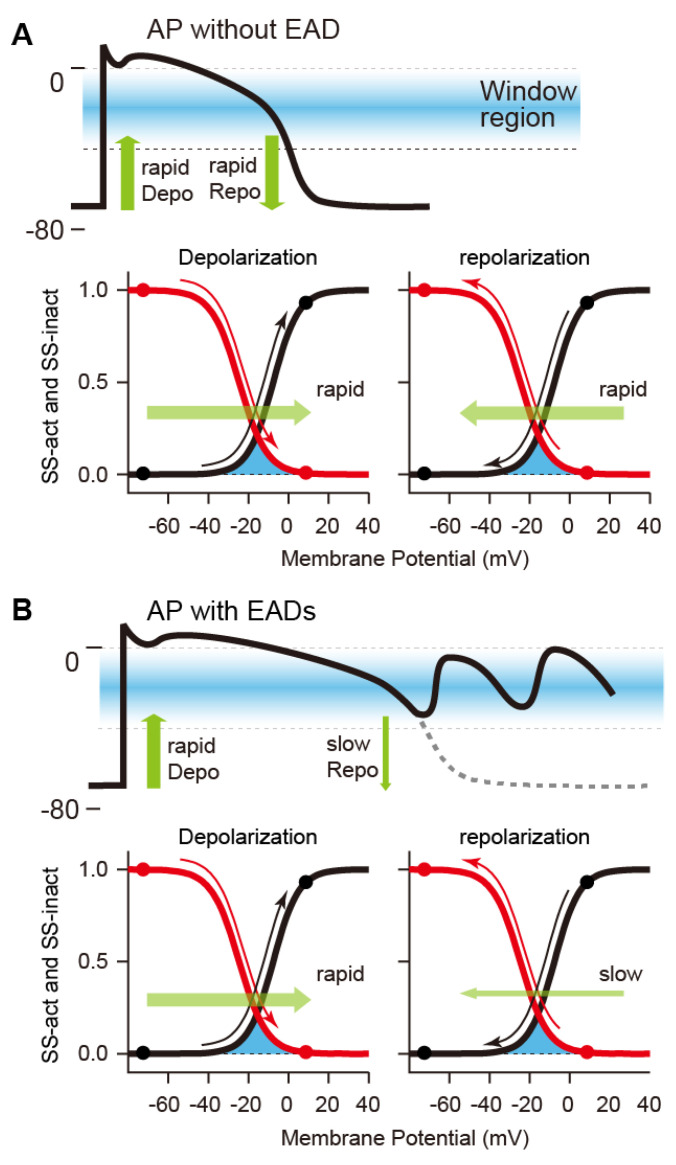
Relationships between membrane potential changes in response to depolarization and repolarization in a normal AP without EAD (**A**) and an abnormal AP with EADs (**B**) and voltage-dependent activation/inactivation of the L-type Ca^2+^ channel. Excessive APD prolongation due to slow repolarization leads to a long-lasting stay of the membrane potential in the L-type Ca^2+^ channel current (*I*_CaL_) window current region (blue areas), resulting in large reactivation of *I*_CaL_. SS-act: steady-state activation curve (black lines); SS-inact: steady-state inactivation curve (red lines); Depo: depolarization; Repo: repolarization.

**Figure 3 biomolecules-12-00459-f003:**
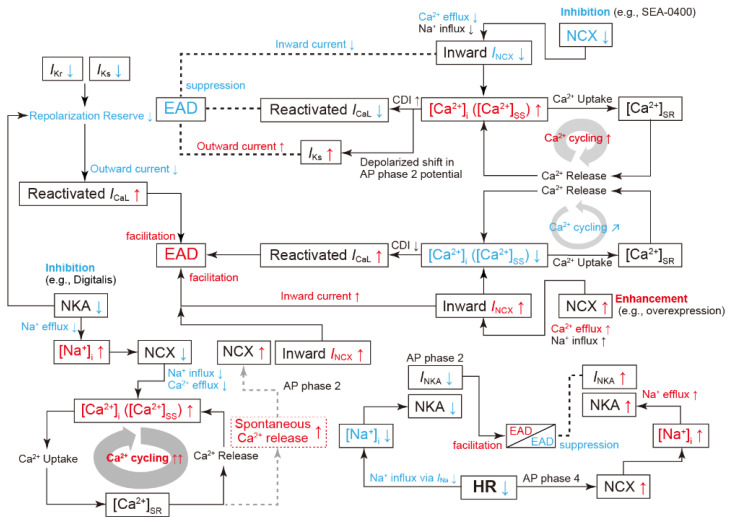
Schematic diagram showing how the dynamics of individual ion channels, transporters, [Ca^2+^]_I_, and [Na^+^]_i_ contribute to EAD generation and regulation. The diagram depicts the major functional components (*I*_Ks_, *I*_Kr_, *I*_CaL_, *I*_NCX_, and *I*_NKA_), factors involved in Ca^2+^- and Na^+^-handling, heart rate (HR), and their interactions related to EAD generation and regulation. The upward and downward arrows represent an increase (or enhancements) and decrease (or attenuations) in each factor, respectively. A decrease in HR (bradycardia) decreases Na^+^ influx via the Na^+^ channel activation, resulting in a decrease in [Na^+^]_i_. This reduction in [Na^+^]_i_ facilitates EAD generation through a decrease in outward *I*_NKA_ during AP phase 2. On the other hand, a decrease in HR prolongs AP phase 4, i.e., diastolic interval (DI), and the prolongation of DI increases the amount of Na^+^ influx through NCX and thus causes [Na^+^]_i_ elevation. This [Na^+^]_i_ elevation increases Na^+^ efflux through NAK and outward *I*_NAK_ during AP phase 2. This may counteract the decrease in [Na^+^]_i_ due to reduced *I*_Na_ and the resulting reduction in *I*_NKA_ (modified from Figure 11 in [85]). For details of other depicted interactions that affect EAD formation, see text.

**Figure 4 biomolecules-12-00459-f004:**
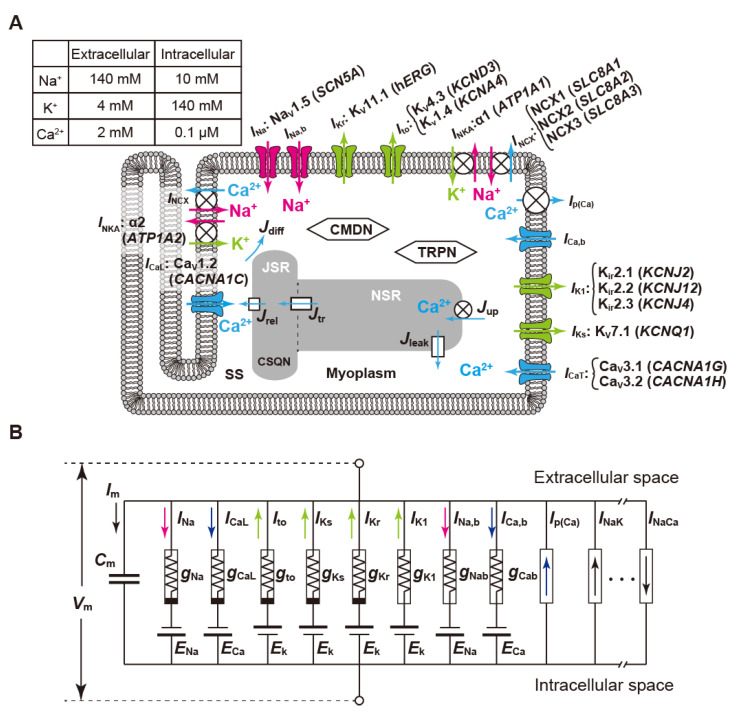
Schematic diagram of the structure and electrical properties of a ventricular myocyte. (**A**): Ion channel and transporter molecules embedded in the cell membrane, sarcoplasmic reticulum (Ca^2+^ cycling), and Ca^2+^-binding molecules related to excitation and contraction. (**B**): An equivalent circuit model of the cell membrane, composed of the ion channel conductance (g), membrane capacitance (*C*_m_), and electromotive force (*E*) representing ion concentration gradients. The current (*I*) flowing through each voltage-gated ion channel is represented by the product of the time-dependent variable conductance and driving force as the difference between the membrane potential (*V*_m_) and individual reversal potential. *g_x_*: maximum conductance of ion channels *x*, for *x* = Na^+^, Ca^2+^, K^+^, etc.; *I*_Na_: fast sodium channel current; *I*_to_: transient outward K^+^ channel current; *I*_Kr_ and *I*_Ks_: fast and slow components, respectively, of delayed rectifier K^+^ channel currents; *I*_CaL_: L-type Ca^2+^ channel current; *I*_CaT_: T-type Ca^2+^ channel current; *I*_K1_: inward rectifier K^+^ channel current; *I*_NKA_: Na^+^/K^+^ ATPase current; *I*_NCX_: Na^+^/Ca^2+^ exchanger current; *I*_p(Ca)_: Ca^2+^ pump current in the sarcolemmal membrane; *J*_rel_: SR Ca^2+^ release flux by Ca^2+^-induced Ca^2+^-release; *J*_up_: Ca^2+^ uptake flux via Ca^2+^ pump (SERCA) in the SR; *J*_leak_: Ca^2+^ leakage flux from the SR; JSR: junctional SR; NSR: network SR; CMDN: calmodulin; TRPN: troponin; SS: a subspace of the myoplasm. For details of other symbols, refer to references [99,114,115,116,117].

**Figure 5 biomolecules-12-00459-f005:**
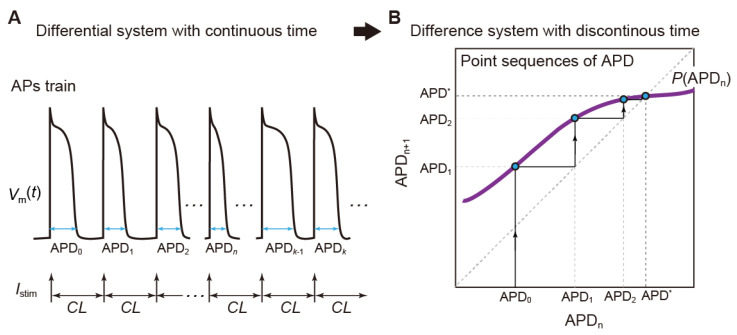
Examples of a dynamical system with continuous time and with discrete time. (**A**): An action potential (AP) train evoked in a cardiomyocyte with stimuli (*I*_stim_) applied at a cycle length of *CL*, which is a typical example of the response of a dynamical system with continuous time. AP duration of the AP evoked by the *i*th stimulus is denoted as APD_i_ for *i* = 0, 1, 2, …, *n*, …, *k* − 1, *k*, …. (**B**): A point sequence consisting of APD values determined for the AP train and plotted on the (APD*_n_*, APD*_n_*_+1_)-plane. If a mapping *P* (Poincaré map) is obtained to represent the relationship between a point (APD*_n_*) and the next point (APD*_n_*_+1_), i.e., the dynamics of the point sequence, then the differential system with continuous time is transformed into a difference system with discrete time. The dynamics of such the difference system with discrete time can then be studied by projecting the point sequence dynamics into the state space, e.g., (APD*_n_*, APD*_n_*_+1_)-plane. Thereby, it is possible to examine the dynamics of the original differential system with continuous time more efficiently. In general, the Poincaré map is difficult to obtain analytically and is mostly obtained numerically. For examples of experimental and numerical methods for obtaining Poincaré maps, see [124,125].

**Figure 6 biomolecules-12-00459-f006:**
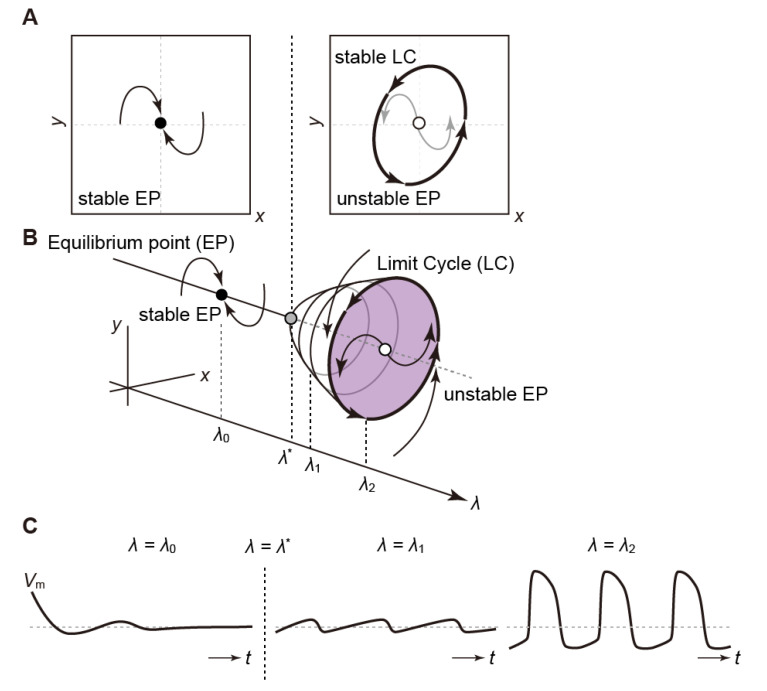
Hopf bifurcation. Examples of the dynamic responses to changes in the parameter *λ* in the 2-dimensional (*x*, *y*)-state space (**A**) and 3-dimensional (*x*, *y*, *λ*)-space (**B**). Schematic examples of membrane potential changes at the parameter λ_0_, λ_1_, λ_2_ in the 3-dimensional (x, y, λ)-space of panel (**A**,**C**). Changing the system parameter *λ*, the stability of an equilibrium point (EP) changes via a Hopf bifurcation. Before and after the occurrence of a Hopf bifurcation (*λ**), a stable EP becomes unstable with the emergence of a limit-cycle (LC) oscillation.

**Figure 7 biomolecules-12-00459-f007:**
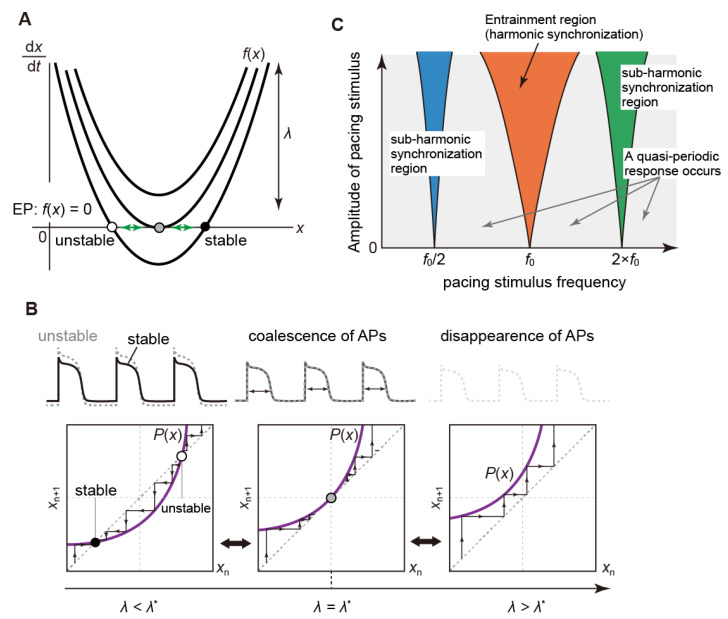
Saddle-node (SN) bifurcation. Schematic diagrams of the SN bifurcations of equilibrium (**A**) and fixed (**B**) points. As simple examples, in the dynamical systems d*x*/d*t* = *f*(*x*, *λ*) (**A**) and *x_n_*_+1_ = *P*(*x_n_*, *λ*) (**B**), the functions *f*(*x*) and *P*(*x_n_*) are varied up and/or down by changing system parameter *λ*. The SN bifurcation of equilibrium points occurs when the curve of *f*(*x*) touches the *x*-axis (d*x*/d*t* = 0 axis). On the other hand, the SN bifurcation of fixed points occurs when the curve of *P*(*x_n_*) touches the diagonal line (*x_n_*_+1_ = *x_n_*) at a value *λ*^*^ of parameters. After that, a node and a saddle point emerge or disappear. A schematic diagram of Arnold’s tongue structure is also shown for a non-autonomous system with stimuli of various amplitudes and frequencies (**C**). Colored regions indicate parameter regions in which a periodic oscillation observed in an autonomous system can be entrained by external periodic stimuli, e.g., the orange region represents harmonic synchronized oscillation observed in the non-autonomous system. In general, the harmonic synchronized region and even the sub-harmonic synchronous region (e.g., blue and green regions) are divided in the parameter space by SN bifurcation sets. Oscillations that are asynchronous to the periodic stimuli, such as quasi-periodic oscillations, appear when set to parameters outside the synchronization region. *f*_0_: an intrinsic frequency of the limit-cycle oscillation in the autonomous system.

**Figure 8 biomolecules-12-00459-f008:**
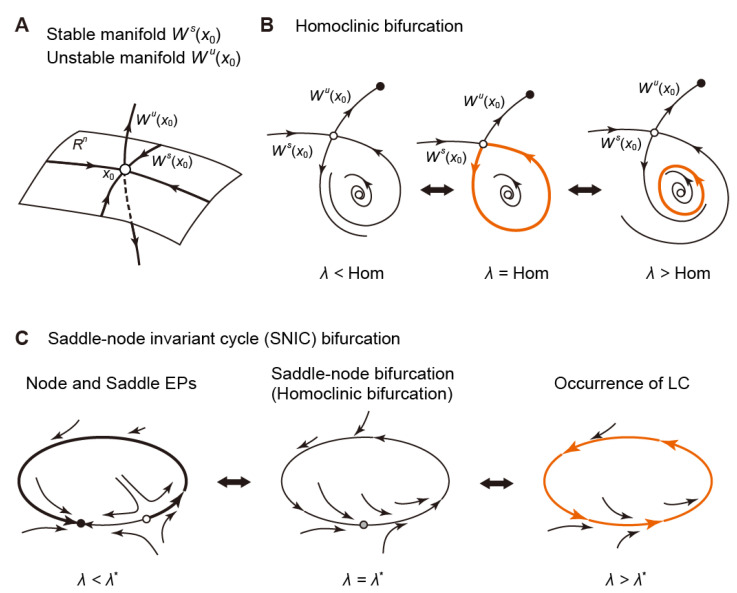
Bifurcation phenomena associated with stable and unstable manifolds of an equilibrium point. (**A**) A schema of stable and unstable manifolds of the equilibrium point *x*_0_ in the state space. *W*^s^(*x*_0_), stable manifold; *W*^u^(*x*_0_), unstable manifold. (**B**) Homoclinic bifurcation. When the parameter *λ* becomes the value of Hom, the stable and unstable manifolds of the equilibrium point *x*_0_ coincide, giving rise to a closed-loop orbit, which is referred to as “homoclinic orbit”. This closed-loop orbit is vulnerable to changes in the parameters and breaks down immediately. (**C**) Saddle-node invariant cycle bifurcation. At the bifurcation parameter value *λ**, saddle-node and homoclinic bifurcations occur simultaneously.

**Figure 9 biomolecules-12-00459-f009:**
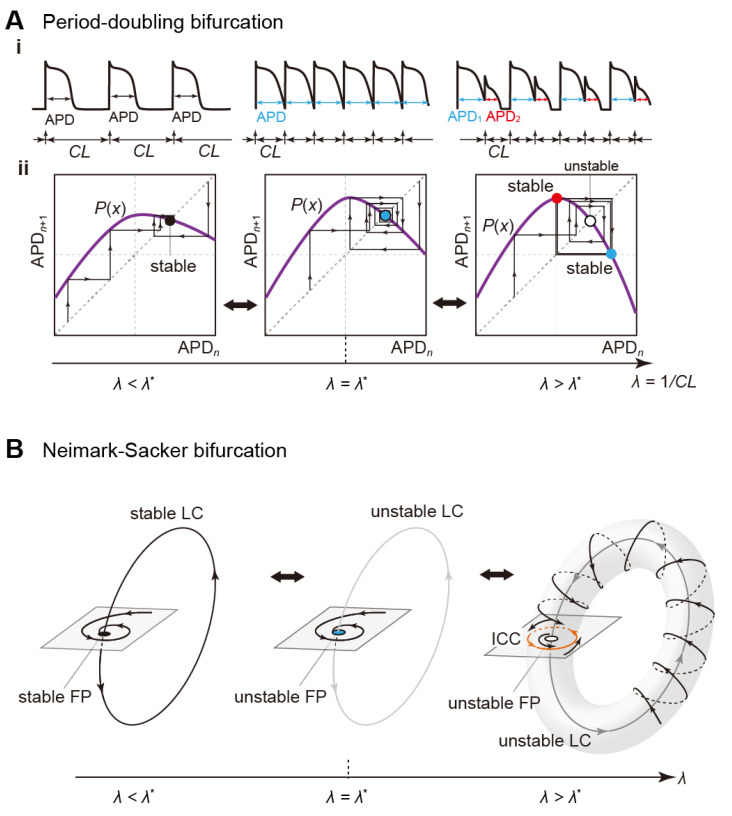
Schematic diagrams of period-doubling and Neimark–Sacker bifurcations. (**A**): A typical example of the period-doubling (PD) bifurcation in non-autonomous systems is AP alternans (**i**). As the cycle length (*CL*) becomes shorter, the stable fixed point on the (APD*_n_*, APD*_n_*_+1_)-plane becomes unstable at the period-doubling bifurcation point (*λ**), and a pair of periodic points is generated (**ii**). (**B**): The change in a limit-cycle (LC) oscillation through the Neimark–Sacker (NS) bifurcation in autonomous systems. The NS bifurcation occurs at *λ** as the parameter *λ* changes. Then, the LC oscillation becomes unstable and a torus-like oscillatory response, so-called “quasi-periodic oscillation”, occurs around the unstable LC oscillation. When the quasi-periodic oscillation is discretized as a point sequence via the Poincaré mapping, a closed curve, which is referred to as an “invariant closed curve (ICC)”, appears around the unstable fixed point that reflects the unstable LC oscillation.

**Figure 10 biomolecules-12-00459-f010:**
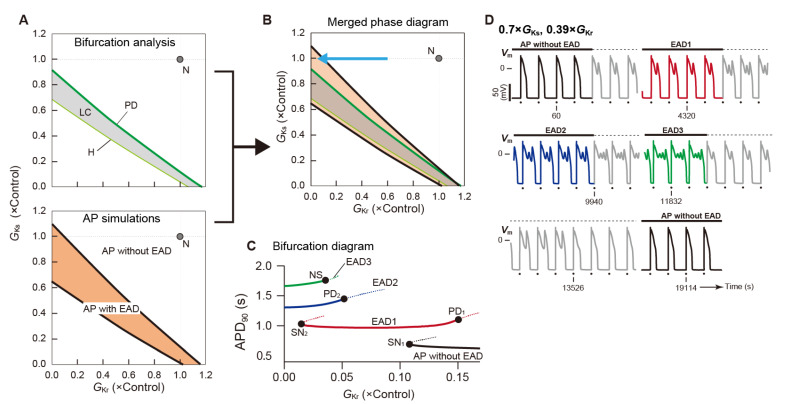
Bifurcation phenomena and action potential behaviors observed in the Kurata model [112] of a human ventricular myocyte. (**A**) A two-parameter bifurcation diagram on the (*G*_Kr_, *G*_Ks_)-parameter plane (**top**) obtained using MATCONT [154] and a phase diagram (**bottom**) obtained by AP simulations. The maximum conductances of the rapid (*G*_Kr_) and slow (*G*_Ks_) components of delayed rectifier K^+^ channel currents are expressed as normalized values, i.e., ratios to the control values. In the two-parameter bifurcation diagram (**top**), the symbols H and PD represent the loci of parameter sets that cause the Hopf bifurcation of an equilibrium point and period-doubling bifurcation of a limit cycle (LC), respectively. The gray region indicates the area of parameters in which a stable LC can be observed. On the other hand, in the phase diagram (**bottom**), the orange and white regions represent parameter regions in which an AP with and without EAD, respectively, can be observed in the Kurata model. The gray point with the symbol *N* indicates the control condition with the normal *G*_Kr_ and *G*_Ks_. (**B**) A merged phase diagram. (**C**) A one-parameter bifurcation diagram of APD at 90% repolarization (APD_90_) in each AP response as a function of *G*_Kr_; see the blue arrow in panel (**B**), which indicates the change in *G*_Kr_ with the fixed normal *G*_Ks_ (1). The solid and dashed lines in (**C**) represent stable and unstable AP responses, respectively. SN: saddle-node bifurcation; EAD1–3: AP with 1–3 EAD(s); PD: period-doubling bifurcation. NS: Neimark–Sacker bifurcation. (**D**) An example of tetra-stable AP dynamics in the Kurata model at 0.39 × *G*_Kr_ with 0.70 × *G*_Ks_. Colored and grey lines indicate the steady-state and transient responses, respectively. Dots indicate the application of current pulses. Pacing cycle length = 2 s. Each panel was modified from [85,130].

**Figure 11 biomolecules-12-00459-f011:**
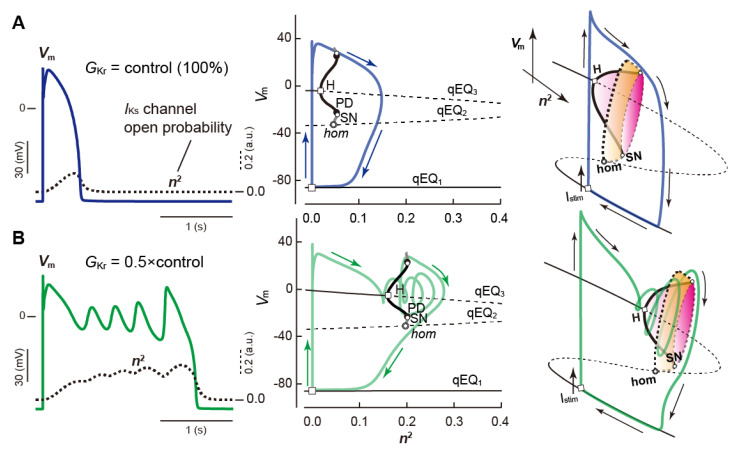
Examples of the slow–fast decomposition analysis. AP simulations and slow–fast decomposition analyses were performed using the Kurata model [112] with *G*_Kr_ = 100% (**A**) and *G*_Kr_ = 0.5 × control (**B**). (**Left**) The membrane potential (*V*_m_) and the time course of *I*_Ks_ open probability (*n*^2^) in the full system. (**Center**) One-parameter bifurcation diagrams of the fast subsystem, depicting quasi-equilibrium potentials (qEQ_1–3_) and the potential extrema of quasi limit cycles (qLCs) as a function of the slow variable *n*^2^. The trajectory of the full system projected onto the (*V*_m_, *n*^2^)-plane is superimposed on the one-parameter bifurcation diagram of the fast subsystem. Solid and dashed thin lines of qEQ_1–3_ indicate stable and unstable equilibrium potentials, respectively. The black and gray thick lines are stable and unstable qLCs, respectively. H: Hopf bifurcation; SN: saddle-node bifurcation of limit cycles; PD: period-doubling bifurcation of limit cycles; hom: homoclinic bifurcation. (**Right**) Schematic diagrams representing the relationships between the dynamic behavior projected onto the 3-dimensional state space of the full system and the bifurcation structure of the fast subsystem. Modified from [197].

## Data Availability

Not applicable.

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
