# Peer review of "Bifurcations and Proarrhythmic Behaviors in Cardiac Electrical Excitations"

_biomolecules, 2022, doi:10.3390/biom12030459_

Round 1

Reviewer 1 Report

The article describes some basic concepts of cardiac electrophysiology and bifurcation theory.  The proarrhythmic behaviors explained using bifurcation analysis only addressed EAD generation. Little is mentioned about reentrant phenomena and delay after depolarization as arrhythmic mechanisms and the potential contribution of bifurcation to these other basic arrhythmic mechanisms. At least a brief description of the limitations should be included.

Reviewer 2 Report

In this paper the authors review the main mechanisms underlying membrane excitability in cardiac cells, describe key concepts of bifurcation theory in dynamical system, and show how they have been applied to cardiac cellular electrophysiology, with particular focus on abnormal membrane repolarization.

I think there is a real need for papers reviewing this field, given the striking effectiveness of the described mathematical tools in capturing the extraordinary complexity of cardiac dynamics.

I also believe the plan of the work is quite nicely conceived, and well documented by references to the literature.

On the other hand, unfortunately, I find the manuscript, particularly the first paragraphs, very badly written, with numerous conceptual mistakes. Though I’m not a mother language, I can tell the English is very poor, with repetitions and misused plurals/singulars, verb forms, repetitions.

I do not make a distinction here between major or minor relevance of my concerns. I believe that all of them have to be faced by the authors before this review could be suitable for publication in Biomolecules.

Title.

Cardiac excitations” should be changed in “cardiac electrical excitation”, or, “cardiac cellular repolarization

Introduction.

Many sentences are vague and inaccurate both in their premises and conclusions (“Dynamics is that states change over time”, “So, the dynamics of a model could be complex”, …etc), other simply wrong. There are at least two wrong or at least very inaccurate definitions of arrhythmias only in this section.

Line 54.

 “Ordinary cardiac muscles”. I would rather say “working cardiac muscle

Lines 88-90.

Sentence is inaccurate. The range of physiological potentials is determined by electrochemical gradient and not vice versa.

Lines 93-94.

This is a quite wrong statement! If the channels remain open, the membrane potential will reach the reversal potential of the permeant ion, and the ion gradient across the membrane will remain the same.

Lines 103-104.

NaK pump is not a carrier                                      

Line 128.

Here it should perhaps be mentioned also the relevant role of reverse mode of NCX in the early phase of cardiac AP and EC coupling.

Line 170.

A little later,” quite vague form

Line 188

LTCCs also become inactive” it is worth mentioning the two, voltage- and calcium-dependent, mechanisms of inactivation.

Line 194.

“After that, IK1channels continue to discharge K+out of the cardiomyocyte, consequently flowing the outward current and maintaining a deep resting membrane potential;” This sentence is wrong. In principle IK1 would maintain the resting potential only with its reversal potential, without need of any current flowing, actually, just because no current is flowing at rest.

Line 249.

The key concept of reserve of repolarization has to be explained and introduced before this discussion.

Line 282.

Spontaneous sarcoplasmic calcium releases are also the recognized mechanism of DADs, which should be mentioned here since the title is “afterdepolarizations”.

Line 317.

I would refer the movement of ions to the reversal potentials, which appear in the electrical circuit, rather than ion concentration gradients, which do not appear.

Line 428-431.

The authors have just defined the working myocardium as non-autonomous, and apply to it the concept of “equilibrium point” developed for autonomous systems.

Line 482.

It would be helpful if the authors could provide examples of numerical and analytical mapping functions. I’m surprised rate-dependence and restitution properties are not explicitly addressed at this time, though they are referenced at [130-133].

Figure 6 and figure 7.

I would recommend to add in these, or in additional figures, sequences of cardiac action potentials undergoing the described bifurcations-types. This is done with efficacy, for example, in figure 5A and 8A.

Line 596-597.

For example, if the APD of cardiomyocytes is constant under a certain CL, then CL shortening leads to DI shortening because of the relational expression of CL = APD + DI.” This sentence, as it is formulated now, is incorrect. If APD is constant at a given CL, say during steady state pacing at constant CL, if CL changes to another constant value, APD would immediately change starting from the first beat at the new CL (electrical restitution), and reach a new and different steady state (steady state rate dependence). I cannot imagine an instance where changes in CL, even only at a beat to beat basis, only affect DI. This point should be clarified.

Lines 636-642.

Many cardiomyocyte models … for questions.” I believe that the need for numerical solutions can be explained better than this.

Line 668.

the points that could not explain in this review.” A “be” is missing.

Round 2

Reviewer 2 Report

The authors have satisfactorily answered all my concerns and made the corresponding changes to the manuscript, which I now judge suitable to be published in Biomolecules.